# Divergent synthesis of *N*-heterocycles via controllable cyclization of azido-diynes catalyzed by copper and gold

Wen-Bo Shen[1], Qing Sun[2], Long Li[1], Xin Liu[1], Bo Zhou[1], Juan-Zhu Yan[1], Xin Lu[2] & Long-Wu Ye [1,3]

Gold-catalyzed intermolecular alkyne oxidation by an N–O bond oxidant has proven to be a powerful method in organic synthesis during the past decade, because this approach would enable readily available alkynes as precursors in generating α-oxo gold carbenes. Among those, gold-catalyzed oxidative cyclization of dialkynes has received particular attention as this chemistry offers great potential to build structurally complex cyclic molecules. However, these alkyne oxidations have been mostly limited to noble metal catalysts, and, to our knowledge, non-noble metal-catalyzed reactions such as diyne oxidations have not been reported. Herein, we disclose a copper-catalyzed oxidative diyne cyclization, allowing the facile synthesis of a wide range of valuable pyrrolo[3,4-*c*]quinolin-1-ones. Interestingly, by employing the same starting materials, the gold-catalyzed cascade cyclization leads to the divergent formation of synthetically useful pyrrolo[2,3-*b*]indoles. Furthermore, the proposed mechanistic rationale for these cascade reactions is strongly supported by both control experiments and theoretical calculations.

[1] Collaborative Innovation Center of Chemistry for Energy Material, State Key Laboratory of Physical Chemistry of Solid Surfaces, and Fujian Provincial Key Laboratory of Chemical Biology, College of Chemistry and Chemical Engineering, Xiamen University, Xiamen 361005, China. [2] Collaborative Innovation Center of Chemistry for Energy Material, State Key Laboratory of Physical Chemistry of Solid Surfaces, and Fujian Provincial Key Laboratory of Theoretical and Computational Chemistry, College of Chemistry and Chemical Engineering, Xiamen University, Xiamen 361005, China. [3] State Key Laboratory of Organometallic Chemistry, Shanghai Institute of Organic Chemistry, Chinese Academy of Sciences, Shanghai 200032, China. Correspondence and requests for materials should be addressed to X.L. (email: xinlu@xmu.edu.cn) or to L.-W.Y. (email: longwuye@xmu.edu.cn)

Highly efficient construction of *N*-heterocycle skeletons is one of the most important themes in organic synthesis. The structurally diverse and interesting family of tricyclic *N*-heterocycles, such as pyrrolo[3,4-*c*]quinolin-1-ones[1–7] and pyrrolo[2,3-*b*]indoles[8–11], are important structural motifs that can be frequently observed in bioactive molecules as well as in natural products (Fig. 1). It is surprising, however, that only a few preparative methods have been reported, with most employing the corresponding quinolines[12–14] and indoles[15–17] as precursors,

respectively. Thus, new synthetic approaches for the direct construction of these skeletons are highly desired, especially those based on the assembly of structures directly from readily available and easily diversified building blocks.

Gold-catalyzed intermolecular alkyne oxidation by an N–O bond oxidant, presumably via an α-oxo gold carbenoid intermediate, has attracted considerable interest during the past decade because this approach would enable readily available and safer alkynes to replace not easily accessible and hazardous α-

**Fig. 1** Selected examples bearing the pyrrolo[3,4-*c*]quinolin-1-one and pyrrolo[2,3-*b*]indole core structure. Some of these molecules are synthesized in the next section

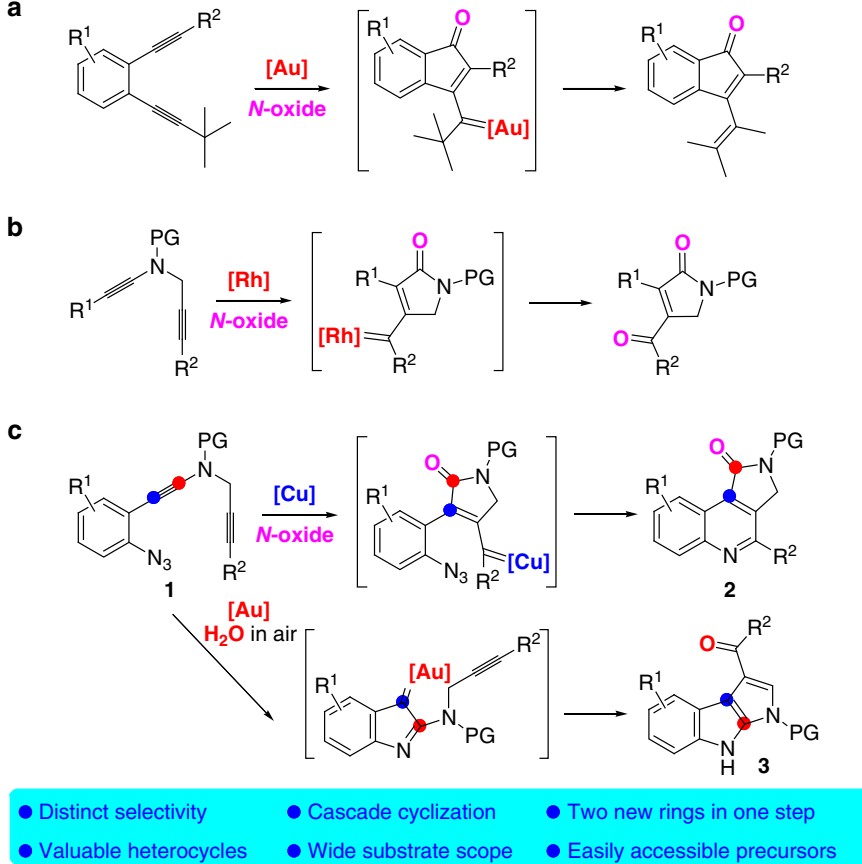

**Fig. 2** Transition-metal-catalyzed oxidative diyne cyclization. **a** Au-catalyzed oxidative diyne cyclization (Hashmi). **b** Rh-catalyzed oxidative diyne cyclization (Tang). **c** Cu-catalyzed oxidative diyne cyclization and Au-catalyzed cascade cyclization (this work)

**Table 1 Optimization of reaction conditions[a]**

| Entry | Catalyst | Oxidant | $T$ (°C) | Yield (%)[b] | |
|---|---|---|---|---|---|
| | | | | **2a** | **3a** |
| 1 | Zn(OTf)$_2$ (10 mol%) | **4a** | 80 | 12 | <1 |
| 2 | In(OTf)$_3$ (10 mol%) | **4a** | 80 | 15 | <1 |
| 3 | Sc(OTf)$_3$ (10 mol%) | **4a** | 80 | 13 | <1 |
| 4 | Y(OTf)$_3$ (10 mol%) | **4a** | 80 | 12 | <1 |
| 5 | Cu(OTf)$_2$ (10 mol%) | **4a** | 80 | 18 | <1 |
| 6 | Cu(CH$_3$CN)$_4$PF$_6$ (10 mol%) | **4a** | 80 | 41 | <1 |
| 7 | [Rh(CO)$_2$Cl]$_2$ (5 mol%) | **4a** | 80 | 6 | <1 |
| 8 | Ph$_3$PAuNTf$_2$ (5 mol%) | **4a** | 80 | 8 | 48 |
| 9 | IPrAuNTf$_2$ (5 mol%) | **4a** | 80 | <3 | 32 |
| 10 | Cu(CH$_3$CN)$_4$PF$_6$ (10 mol%) | **4b** | 80 | 72 | <1 |
| 11[c] | Cu(CH$_3$CN)$_4$PF$_6$ (10 mol%) | **4b** | 60 | 85 | <1 |
| 12 | Ph$_3$PAuNTf$_2$ (5 mol%) | – | 25 | <1 | 58 |
| 13[d] | Ph$_3$PAuNTf$_2$ (5 mol%) | – | 25 | <1 | 86 |
| 14[e] | Ph$_3$PAuNTf$_2$ (2 mol%) | – | 25 | <1 | 87 |

[a]Reaction conditions: the reaction was performed with **1a** (0.1 mmol), **4** (0.15 mmol), and catalyst (2–10 mol%) in DCE (2 mL) at 25–80 °C in vials
[b]Measured by $^1$H NMR using diethyl phthalate as the internal standard
[c]2 h
[d]In CH$_3$NO$_2$, 5 min
[e]In CH$_3$NO$_2$, 30 min

diazo carbonyls as precursors in generating α-oxo metal carbenes[18–30]. Among those, gold-catalyzed oxidative cyclization of dialkynes has received particular attention because this chemistry offers great potential to build structurally complex cyclic molecules[31–35]. For example, Hashmi et al. reported an elegant protocol for the gold-catalyzed oxidative diyne cyclization via a presumable 1,6-carbene transfer (Fig. 2a)[32]. Notable is that haloalkynes are typically required for this strategy. Such a gold-catalyzed oxidative diyne cyclization has also been well exploited in the synthesis of various functionalized O-heterocycles by Zheng and Zhang[33] and Ji et al.[34]. In addition, Tang et al. disclosed that rhodium could also catalyze this type of diyne oxidation (Fig. 2b)[35]. Despite these significant achievements, these alkyne oxidations have been mostly limited to noble metal catalysts, and, to our knowledge, non-noble metal-catalyzed such as diyne oxidation has not been reported.

Inspired by our recent study on ynamide chemistry[36–43], we envisioned that the synthesis of pyrrolo[3,4-c]quinolin-1-ones **2** might be accessed through such an oxidative cyclization of N-propargyl (azido)ynamides **1**. However, realizing this cascade reaction is highly challenging because of two competing reactions. First, the generated vinyl metal carbene is highly reactive and often suffers the overoxidation by the same oxidant[32, 35, 41, 42], in addition to many other side reactions. Second, the azido group would be expected to attack the ynamide directly to initiate the relevant alkyne amination via a presumable α-imino metal carbene pathway[44–55]. Herein, we describe the realization of a copper-catalyzed oxidative diyne cyclization protocol that allows the facile synthesis of a variety of valuable pyrrolo[3,4-c]quinolin-1-ones. Furthermore, by employing the same starting materials, the gold-catalyzed cascade cyclization leads to the divergent

formation of pyrrolo[2,3-b]indoles. In addition, the mechanistic rationale for these cascade reactions, in particular accounting for the distinct selectivity, is also well supported by density functional theory (DFT) calculations.

## Results

**Optimization of reaction conditions**. Table 1 shows the realization of the cascade cyclization of ynamide **1a** in the presence of various transition metals (for more details see Supplementary Table 1, Supporting Information. To our delight, the tandem reaction indeed produced the desired pyrrolo[3,4-c]quinolin-1-one **2a** under the previously optimized reaction conditions[42], albeit in low yield (Table 1, entry 1). We then investigated other non-noble metals (Table 1, entries 2–6), and were pleased to find that Cu(CH$_3$CN)$_4$PF$_6$ catalyzed the oxidative cyclization to produce the desired **2a** in 41% yield (Table 1, entry 6). Of note, rhodium (Table 1, entry 7)[35] and Brønsted acids[56–59] such as TsOH and TfOH were not effective in promoting this reaction (for more details see Supplementary Table 1). Interestingly, pyrrolo[2,3-b]indole **3a** was obtained as the main product in the presence of typical gold catalysts such as Ph$_3$PAuNTf$_2$ and IPrAuNTf$_2$ (Table 1, entries 8 and 9). Further screening of oxidants revealed that the use of quinoline N-oxide **4b** led to a significantly improved yield in the presence of Cu(CH$_3$CN)$_4$PF$_6$ as catalyst (Table 1, entry 10, for more details see Supplementary Table 1), and **2a** could be formed in 85% yield at 60 °C (Table 1, entry 11). In addition, condition optimization on the formation of **3a** was also explored (for more details see Supplementary Table 1), and it was found that slightly improved yield was obtained by employing Ph$_3$PAuNTf$_2$ as catalyst in the absence of

**Fig. 3** Reaction scope for the formation of pyrrolo[3,4-c]quinolin-1-ones **2**. Reaction conditions: [**1**] = 0.05 M; yields are those for the isolated products

**Fig. 4** Copper-catalyzed oxidative cyclization of chiral N-propargyl (azido) ynamides **1**. Substrate scope of chiral N-propargyl ynamides **1**

oxidant (Table 1, entry 12). Gratifyingly, 86% yield was achieved by using $CH_3NO_2$ as solvent (Table 1, entry 13), and similar yield was obtained when the catalyst loading was reduced to 2 mol% (Table 1, entry 14). Notably, no formation of **3a** was observed under copper catalysis (for more details see Supplementary Table 1).

**Synthesis of pyrrolo[3,4-c]quinolin-1-ones via Cu catalysis.** With the optimal reaction conditions in hand (Table 1, entry 11), the reaction scope of the copper-catalyzed synthesis of pyrrolo [3,4-c]quinolin-1-ones was then explored (Fig. 3). The reaction proceeded smoothly with different aryl-substituted ynamides ($R^2$ = Ar), affording the desired γ-lactam-fused quinolines **2a–h** in generally good to excellent yields (Fig. 3, entries 1–8, **2a** was

confirmed by X-ray diffraction, for more details see Supplementary Table 2). In addition, heterocycle-substituted ynamide **1i** was also a suitable substrate for this oxidative cyclization to produce the corresponding **2i** in a serviceable yield (Fig. 3, entry 9), whereas none of the desired **2j** was observed with alkyl-substituted ynamide **1j** (Fig. 3, entry 10). The method worked efficiently for various aryl-substituted ynamides bearing both electron-donating and -withdrawing groups, and the desired **2k–o** were obtained in 63–94% yields (Fig. 3, entries 11–15). Ynamides containing other protecting groups also reacted well to afford the tricyclic N-heterocycles in 68–85% yields (Fig. 3, entries 16–18). Importantly, no diketone formation via double oxidation by the same oxidant was observed in all cases[32, 35].

This reaction was also extended to substituted N-propargyl ynamides and these chiral substrates could be readily prepared with excellent enantiomeric excesses by using Ellman's tert-butylsulfinimine chemistry (for more details see Supplementary Fig. 87). Thus, the desired enantioenriched tricyclic N-heterocycles **2s–t** were formed in good yields with well-maintained enantioselectivity by employing 8-isopropylquinoline N-oxide **4c** as oxidant (Fig. 4).

**Synthesis of pyrrolo[2,3-b]indoles via Au catalysis.** We also investigated the substrate scope for the gold-catalyzed synthesis of pyrrolo[2,3-b]indoles with the same ynamide substrates under the optimal reaction conditions (Table 1, entry 14). As shown in

**Fig. 5** Reaction scope for the formation of pyrrolo[2,3-*b*]indoles **3**. Reaction conditions: [**1**] = 0.05 M; yields are those for the isolated products

Fig. 5, this alkyne amination-initiated tandem reaction[39, 44–55] proceeded very well and afforded the desired pyrrole-fused indoles **3a–h** in mostly good to excellent yields (Fig. 5, entries 1–8, **3a** was confirmed by X-ray diffraction, for more details see Supplementary Table 3). This chemistry could also be extended to heterocycle- or alkyl-substituted ynamides, leading to the corresponding **3i** and **3j** in 73% and 86% yields, respectively (Fig. 5, entries 9 and 10). Ynamides bearing different aryl groups and protecting groups were also suitable substrates for this gold catalysis to furnish the desired fused *N*-heterocycles in 56–86% yields (Fig. 5, entries 11–18).

Further synthetic transformations of the as-synthesized tricyclic *N*-heterocycles were also explored (Fig. 6). For example, the Ts group in γ-lactam-fused quinoline **2a**, obtained on a gram scale in 77% yield, was easily removed by the treatment with H₂SO₄ to afford the corresponding **5a** in 74% yield, which could be further transformed into DHODH inhibitor **5b**[6]. Alternatively, **5a** could be converted into pyrrolo[3,4-*c*]quinoline-1,3-dione **5c**, known for antibacterial activity against Gram-positive and Gram-negative bacteria[2], via a facile K₂CO₃-mediated air oxidation[60, 61] and metal-free oxidative arene imidation[62]. By using a similar strategy, the synthesis of caspase-3 inhibitor **5d** was achieved starting from the corresponding ynamide **1u**[5]. In addition, pyrrole-fused indole **3a** could be subjected to removal of the Ts

group by NaOH or reduction of the carbonyl group by LiAlH₄ to produce the desired **6a** and **6b**, respectively.

**Mechanistic investigations**. To understand the mechanism of these cyclizations, several control experiments were first conducted. As shown in Fig. 7, control experiments with H₂¹⁸O and ¹⁸O₂ isotopic labeling proved that the oxygen atom in the carbonyl group of **3a** originates from water but not molecular oxygen. Of note, no incorporation of ¹⁸O into the **3a** was observed when **3a** was subjected to the reaction conditions with H₂¹⁸O (for more details see Supplementary Fig. 81).

In addition, when ynamide **1v** was subjected to this copper-catalyzed cascade reaction, no **2v** formation was observed, and the corresponding **2va** was obtained in 66% yield instead (Fig. 8). These results suggested that vinyl copper carbene intermediate was presumably involved in such a diyne oxidation.

Based on the above experimental observations (for more details see Supplementary Figs. 80–85), previously published results[32, 35, 44–55], and on DFT computations (for more details see Supplementary Figs. 75–79) plausible mechanisms for the divergent Cu^I/Au^I-catalyzed synthesis of **2a** and **3a** are illustrated in Fig. 9. First, the catalytic [M^I]-species is preferentially bound to the amide-neighbored, electron-richer triple bond of **1a**, forming precursor **A** (for more details see Supplementary Fig. 75). In the

**Fig. 6** Synthetic applications. **a** Synthesis of bioactive molecules **5b** and **5c**. **b** Synthesis of caspase-3 inhibitor **5d**. **c** Transformation of **3a** into **6a** and **6b**

oxidant-free cycle (for more details see Supplementary Fig. 76), intramolecular cyclization is thus triggered by nucleophilic attack of the proximal N atom of azide to form intermediate **B**, followed by elimination of $N_2$ to form metal-carbenoid intermediate **C**, and a second cyclization to the enylium-cationic intermediate **D**. The latter can readily react with ambient $H_2O$, leading eventually to product **3a** (for more details see Supplementary Figs. 80, 81). The overall barrier height (OBH) (for more details see Supplementary Figs. 78, 79) is determined by the relative free energy of transition state **TSc**, which amounts up to 25.6 kcal/mol in $Cu^I$ catalysis, 9.5 kcal/mol higher than that in $Au^I$ catalysis. This accounts well for the much higher efficiency of the $Au^I$-catalyst in the oxidant-free synthesis of **3a**. In the oxidant-involving cycle (for more details see Supplementary Fig. 77), precursor **A** subjects to nucleophilic attack of oxidant **4a** to form vinyl metal intermediate **B′**. Upon N–O bond cleavage, **B′** transforms into α-oxo metal-carbenoid intermediate **C′** (for details, see the Supporting Information)[63–65], leading smoothly to the final product **2a**[66–70]. It appears that the OBH (for more details see Supplementary Figs. 78, 79) of such oxidant-involving cycle is determined by the relative free energy of transition state **TS_B′**, which amounts up to 18.0 and 23.4 kcal/mol in the $Au^I$- and $Cu^I$-catalyses, respectively. Note that in the presence of oxidant, the oxidant-free cycle may even be favored over the oxidant-involving path, if the former has a lower OBH than the latter. This is true for the $Au^I$-catalyst system, but not true for the $Cu^I$-catalyst system. Accordingly, the oxidant-involving $Cu^I$- and $Au^I$-catalyst systems prefer to produce **2a** and **3a**, respectively (for details, see Supplementary Data 1).

## Discussion

In summary, we have developed a copper-catalyzed oxidative cyclization of azido-diynes, affording a wide range of functionalized pyrrolo[3,4-c]quinolin-1-ones in mostly good to excellent

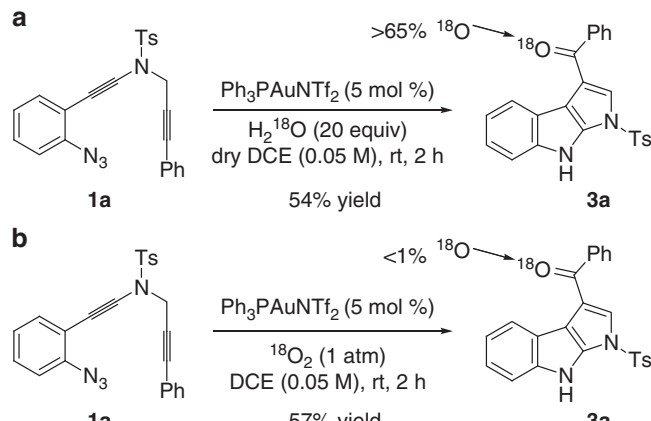

**Fig. 7** Control experiments with $^{18}O$ labeling study. **a** Reactions were run in the presence of 20 equiv of $H_2^{18}O$. **b** Reactions were run in the presence of $^{18}O_2$ atmosphere (1 atm)

yields. Importantly, this protocol represents a non-noble metal-catalyzed diyne oxidation by an N–O bond oxidant. In addition, the gold-catalyzed cascade cyclization of the same substrates leads to the efficient formation of pyrrolo[2,3-b]indoles. Thus, this controllable cascade cyclization enables the efficient and divergent synthesis of two types of valuable tricyclic N-heterocycles from identical starting materials under exceptionally mild conditions. Moreover, the computational study provides further evidence on the feasibility of the proposed mechanism of these cascade reactions, especially for the distinct selectivity. Further studies on other controllable cascade cyclizations are currently underway.

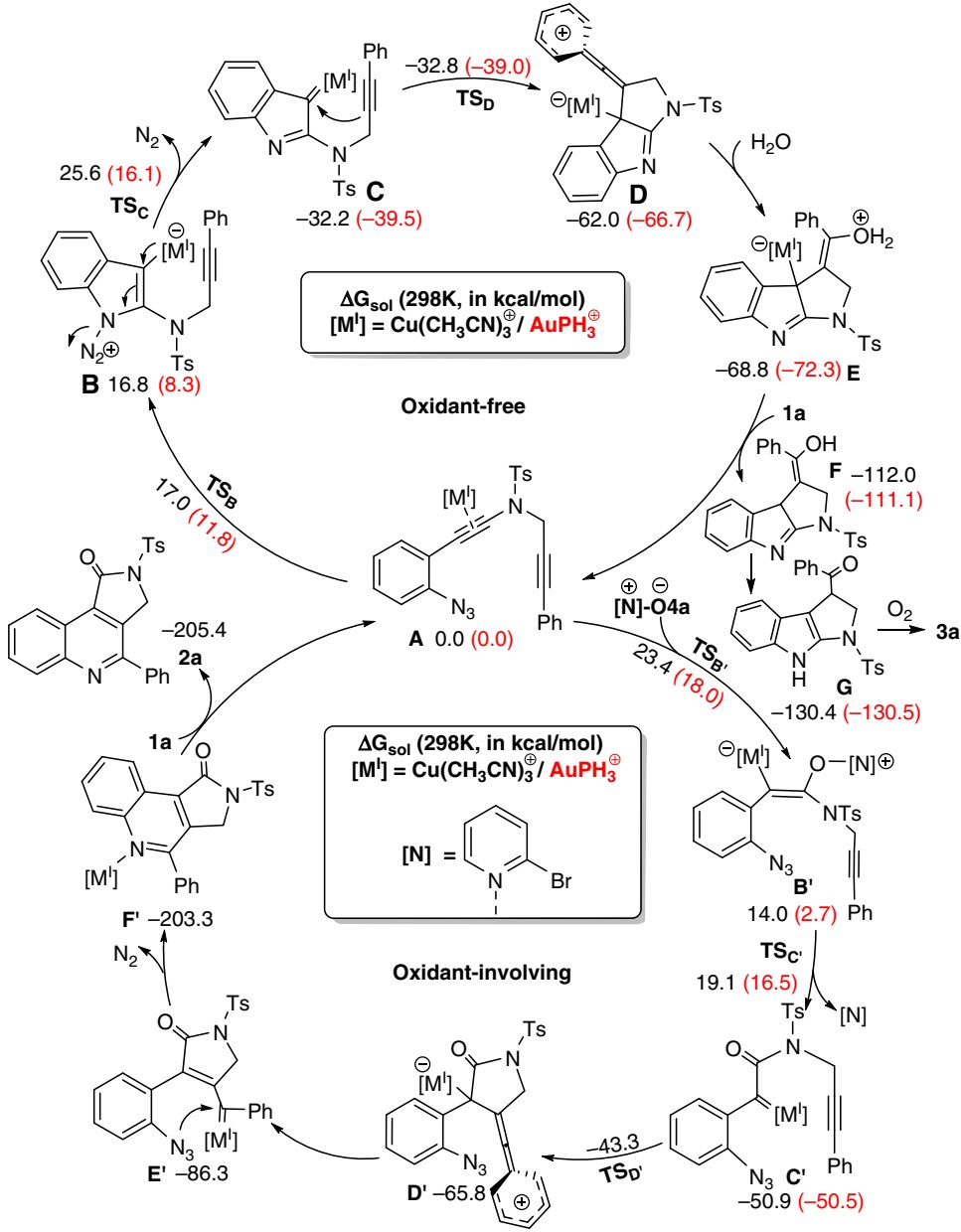

**Fig. 8** Trapping of the presumable vinyl copper carbene intermediate. Substrate scope of alkyl-substituted *N*-propargyl ynamide **1v**

**Fig. 9** Plausible mechanism accounting for the divergent Cu$^I$/Au$^I$-catalyzed formation of **2a/3a**. Relative free energies of key intermediates and transition states were computed at the SMD-M06/DZP level of theory in solvent (DCE for Cu$^I$ catalysis and CH$_3$NO$_2$ for Au$^I$ catalysis) at 298 K. Data for Au$^I$ catalysis were given in parentheses

## Methods

**Materials.** Unless otherwise noted, materials were obtained commercially and used without further purification. All the solvents were treated according to general methods. Flash column chromatography was performed over silica gel (300–400 mesh). See Supplementary Methods for experimental details.

**General methods.** [1]H NMR spectra and [13]C NMR spectra were recorded on a Bruker AV-400 spectrometer and a Bruker AV-500 spectrometer in chloroform-d₃. For [1]H NMR spectra, chemical shifts are reported in ppm with the internal TMS signal at 0.0 ppm as a standard. For [13]C NMR spectra, chemical shifts are reported in ppm with the internal chloroform signal at 77.0 ppm as a standard. Infrared spectra were recorded on a Nicolet AVATER FTIR330 spectrometer as thin film and are reported in reciprocal centimeter (cm$^{-1}$). Mass spectra were recorded with Micromass QTOF2 Quadrupole/Time-of-Flight tandem mass spectrometer using electron spray ionization. [1]H NMR, [13]C NMR, and high-performance liquid chromatography (HPLC) spectra (for chiral compounds) are supplied for all compounds: see Supplementary Figs. 1–74. See Supplementary Methods for the characterization data of compounds not listed in this part.

**General procedure for the synthesis of pyrrolo[3,4-c]quinolin-1-ones 2.** Methylquinoline *N*-oxide (0.3 mmol, 47.7 mg) and Cu(CH₃CN)₄PF₆ (0.02 mmol, 7.5 mg) were added in this order to the ynamide **1** (0.20 mmol) in DCE (4.0 mL) at room temperature. The reaction mixture was stirred at 60 °C and the progress of the reaction was monitored by TLC. The reaction typically took 2 h. Upon completion, the mixture was then concentrated and the residue was purified by chromatography on silica gel (eluent: petroleum ether/dichloromethane) to afford the desired pyrrolo[3,4-c]quinolin-1-one **2**.

**General procedure for the synthesis of pyrrolo[2,3-b]indoles 3.** Ph₃PAuNTf₂ (0.004 mmol, 3.0 mg) was added in this order to the ynamide **1** (0.20 mmol) in CH₃NO₂ (4.0 mL) at room temperature. The reaction mixture was stirred at room temperature and the progress of the reaction was monitored by TLC. The reaction typically took 30 min. Upon completion, the mixture was then concentrated and the residue was purified by chromatography on silica gel (eluent: petroleum ether/ethyl acetate) to afford the desired pyrrolo[2,3-b]indole **3**.

**Data availability.** The X-ray crystallographic coordinates for structures reported in this article have been deposited at the Cambridge Crystallographic Data Centre (CCDC), under deposition number CCDC 1535333 (**2a**) and CCDC 1535335 (**3a**). The data can be obtained free of charge from The Cambridge Crystallographic Data Centre via http://www.ccdc.cam.ac.uk/ data_request/cif. Any further relevant data are available from the authors upon reasonable request.

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

## Acknowledgements

We are grateful for the financial support from the National Natural Science Foundation of China (21572186, 21622204, and 91545105), the Natural Science Foundation of Fujian Province for Distinguished Young Scholars (2015J06003), the President Research Funds from Xiamen University (20720150045), XMU Training Program of Innovation and Enterprenuership for Undergraduates (2016G10384076), and NFFTBS (J1310024). We also thank Professor Dr Nanfeng Zheng from Xiamen University for assistance with X-ray crystallographic analysis.

## Author contributions

W.-B.S., L.L., X.L. and B.Z. performed experiments. Q.S. and X.L. performed DFT calculations. X.L. revised the paper. L.-W.Y. conceived and directed the project and wrote the paper. All authors discussed the results and commented on the manuscript.

## Additional information

**Competing interests:** The authors declare no competing financial interests.

