## [Peer Review File · Nature Communications]

Reviewer #1 (Remarks to the Author):

The merit of this work is to demonstrate the utility of two novel cyclizations catalyzed by two different metal catalysts. The use of Cu catalysts enables formation of important azacyclic core structures whereas a switch to gold catalysts affords different azacycles that are also important cores. The utility of these reactions is well demonstrated. The two reactions proceed via two different mechanisms depending the presence or absence of N-oxidants. In my opinion, the authors report two new reactions to afford useful azacyclic products. Recommendation of this work in *Natural Communications* is appropriate because of significance and novel chemistry. Several minor points are noted before the publications

(1) In Table 1, The use of Cu(CH₃CN)₄PF₆ in the absence of pyridine-oxides should be performed.

(2) This reviewer note the effects of solvents. The authors should check the Au reactions in MeNO₂.

(3) In the reaction mechanism, metal allenyl species D' is somewhat unusual. Did the authors use the calculations to examine the possibility of an alkyne cyclopropanation? This process is better calculated because it is well-accepted in gold chemistry.

(4) The calculations of Cu(I) and Au(I) gold catalysts to distinguish their different activities. The readers will have difficulty to see their active role. In Figure 7, the authors might should additional energy wells for the two different metals so that the readers can read very quickly.

Reviewer #2 (Remarks to the Author):

In this manuscript the authors have reported a cascade cyclization of azido-diyne catalyzed by copper and gold which gave different product. The copper-catalyzed cascade azido-diyne cyclization, which was the first non-noble metal-catalyzed oxidative diyne cyclization, was carried out under mild conditions to form a series of pyrrolo[3,4-c]quinolin-1-ones. The cascade cyclization was involved two cyclizations, the oxidative diyne cyclization and a cyclization by metal-carbenoid. The reaction mechanism was studied by a no-cascade cyclization of ynamide 1v, the second cyclization was prevented owing to the larger steric hindrance metal-carbenoid with lower reactivity, so only the oxidative diyne cyclization occurred to form a eno-pyrrolidone 2va. Furthermore, those copper-catalyzed reactions was maintained the optical activities which was checked from the chiral precursors. Those copper-catalyzed cyclization was also applied to synthesis some useful compound with biological activities. Interestingly, the azido-diyne precursors could form the other N-heterocycle skeletons -- pyrrolo[2,3-b]indoles, which was catalyzed by gold in CH₃NO₂. In these reactions, the azido-diyne also undergo two cyclizations to form enylium-cationic intermediates D. the intermediates D was hydrolyzed and subsequent oxidized to final products. This mechanism was investigated by isotopic labelling experiment which illustrated the oxygen was origin from H₂O. Most of the experiments were well performed. The manuscript fits the scope of nature communications as an article. The work should be acceptable for publication after following issues are addressed.

1. In optimization of reaction conditions, the yield of the product was monitored by NMR, and the recovered starting materials (RSM) was not mentioned. So the reviewer suggest that the yield of product and RSM need to remeasured by HPLC, which could gave more exacter data.

2. It is very interesting the azido-diyne could approach to different product by different catalyst. As the author depicted, the copper-catalyzed process was firstly undergo an oxidative process of [MI]-species, but the [MI]-species was not oxidized for gold-catalyzed process. Maybe the redox potential of Cu[I]-species and Au[I]-species could gave some insight to the reaction mechanisms, so some other explanations for the interesting dual cascade cyclization was needed.

3. The authors adopted different basis sets (LanL2DZ for Cu and SDD for Au) for different transition metals in the same reaction system. However, in the computations, different basis sets

will lead to different values of energy barriers. Therefore, it is not reliable to explain the much higher efficiency of the Au-involved reaction by using the much lower overall barrier height (OBH) with the Au catalyst relative to that with the Cu catalyst. Moreover, it is not convincing to adopt LanL2DZ basis set for Cu based on cost-effective consideration, since the atomic number of copper is smaller than that of gold. Therefore, I suggest the authors use SDD basis set for both copper and gold to perform the computations.

4. In the computational methods part of the Supporting Information, the sentence "The geometries of all the species were fully optimized by using density functional theory (DFT) of the M06 method" is not properly expressed, since M06 is one of the functionals of the density functional theory (DFT).

Reviewer #3 (Remarks to the Author):

Ye, Lu and co-workers report an interesting and important work on the divergent pathway of Cu- and gold-catalyzed cascade cyclization of azido-diyne. Using the same starting materials, the Cu-catalyzed oxidative transformation led to pyrrolo-quinolin-1-ones, while the gold-catalyzed cascade cyclization gave pyrrolo-indoles. The authors also propose the plausible mechanisms for the transformations based on the several control experiments and DFT computations. Although the results are interesting and important, the manuscript needs to undergo a minor revision before it can be recommended for acceptance in Nat.Comm.

Following comments need to be addressed:

- 1) The divergent pathway in Cu- and gold-catalyzed such transformation is quite interesting. However, for the reaction condition of Au/oxidant-involving and Cu/oxidant-free, the authors only investigate the reactivity of substrate 1a having a phenyl group. How about the reactivity of the substrates where the phenyl group in 1a is replaced by an alkyl group, such as 1j, under the condition of Au/oxidant-involving or Cu/oxidant-free? The authors should try the reactions, which maybe is helpful to better understand the mechanism.
- 2) Line 87~93: The authors should describe the reactions with mention of the catalyst involved in these reactions. This would improve the clarity for the readers.

廈門大學
XIAMEN UNIVERSITY
Xiamen, Fujian, China

September 8, 2017

1. Response to comment (reviewer 1):

Comments:

The merit of this work is to demonstrate the utility of two novel cyclizations catalyzed by two different metal catalysts. The use of Cu catalysts enables formation of important azacyclic core structures whereas a switch to gold catalysts affords different azacycles that are also important cores. The utility of these reactions is well demonstrated. The two reactions proceed via two different mechanisms depending the presence or absence of N-oxidants. In my opinion, the authors report two new reactions to afford useful azacyclic products. Recommendation of this work in Natural Communications is appropriate because of significance and novel chemistry.

Several minor points are noted before the publications:

In Table 1, The use of $\text{Cu}(\text{CH}_3\text{CN})_4\text{PF}_6$ in the absence of pyridine-oxides should be performed.

- We first thank the reviewer for her/his kind comments and recommendation.
- We did perform the suggested reaction, and found that $\text{Cu}(\text{CH}_3\text{CN})_4\text{PF}_6$ could not catalyze this cascade cyclization reaction in the absence of an oxidant. This result was actually shown on page 75 (Supplementary Table 1, entry 19) of SI.

2. Response to comment (reviewer 1): This reviewer note the effects of solvents. The authors should check the Au reactions in MeNO_2 .

- We checked the Au-catalyzed cascade cyclization, and found that the reaction proceeded much better in MeNO_2 than in DCE (86% vs 58%).

3. Response to comment (reviewer 1): In the reaction mechanism, metal allenyl species D' is somewhat unusual. Did the authors use the calculations to examine the possibility of an alkyne cyclopropanation? This process is better calculated because it is well-accepted in gold chemistry.

廈門大學

XIAMEN UNIVERSITY

Xiamen, Fujian, China

- We agree that alkyne cyclopropanation (ACP) is well-accepted in gold chemistry. However, we should mention that in addition to ACP, formation of metal allenyl intermediate is not unusual in gold-chemistry, as was proposed in refs. 33 and 34. Actually we did have computationally examined the pathway of alkyne cyclopropanation as an alternative for that of metal allenyl intermediate, but failed to locate the ACP transition state. It is likely due to the large steric repulsion between the two phenyl groups near the approaching metal-carbenoid moiety and the alkyne group that prohibits the ACP pathway.

4. Response to comment (reviewer 1): The calculations of Cu(I) and Au(I) gold catalysts to distinguish their different activities. The readers will have difficulty to see their active role. In Figure 7, the authors might should additional energy wells for the two different metals so that the readers can read very quickly.

- We agree that the energetics given in Fig.7 may not be too reader-friendly, especially for the understanding of the chemoselectivity of gold-catalyzed reaction. We added the Supplementary Figs. 78 and 79 in the SI to show explicitly the energy wells/barriers of the competing pathways for both Au- and Cu-catalyzed reactions.

5. Response to comment (reviewer 2):

Comments:

In this manuscript the authors have reported a cascade cyclization of azido-diyne catalyzed by copper and gold which gave different product. The copper-catalyzed cascade azido-diyne cyclization, which was the first non-noble metal-catalyzed oxidative diyne cyclization, was carried out under mild conditions to form a series of pyrrolo[3,4-c]quinolin-1-ones. The cascade cyclization was involved two cyclizations, the oxidative diyne cyclization and a cyclization by metal-carbenoid. The reaction mechanism was studied by a no-cascade cyclization of ynamide 1v, the second cyclization was prevented owing to the larger steric hindrance metal-carbenoid with lower reactivity, so only the oxidative diyne cyclization occurred to form a eno-pyrrolidone 2va. Furthermore, those copper-catalyzed reactions was maintained the optical activities which was checked from the chiral precursors. Those copper-catalyzed cyclization was also applied to synthesis some useful compound with biological activities. Interestingly, the azido-diyne precursors could form the other N-heterocycle skeletons -- pyrrolo[2,3-b]indoles, which was catalyzed by gold in CH₃NO₂. In these reactions, the azido-diyne also undergo two cyclizations to form enylium-cationic intermediates D. the intermediates D was hydrolyzed and subsequent oxidized to final products. This mechanism was investigated by isotopic labelling experiment which illustrated the oxygen was origin from H₂O. Most of the experiments were well performed. The manuscript fits the scope of nature communications as an article.

The work should be acceptable for publication after following issues are addressed:

廈門大學

XIAMEN UNIVERSITY

Xiamen, Fujian, China

In optimization of reaction conditions, the yield of the product was monitored by NMR, and the recovered starting materials (RSM) was not mentioned. So the reviewer suggest that the yield of product and RSM need to remeasured by HPLC, which could gave more exacter data.

- We thank the reviewer very much for giving us valuable suggestions.
- Actually, no starting material was recovered in all the cases shown in Table 1, as determined by the crude ^1H NMR (also confirmed by TLC). For other cases in reaction condition studies, the recovered starting materials (RSM) were clearly mentioned in Supplementary Table 1 (page 75), also determined by the crude ^1H NMR. Therefore, we don't think it is necessary to remeasure the yield of product and RSM by HPLC.

6. Response to comment (reviewer 2): It is very interesting the azido-diyne could approach to different product by different catalyst. As the author depicted, the copper-catalyzed process was firstly undergo an oxidative process of [MI]-species, but the [MI]-species was not oxidized for gold-catalyzed process. Maybe the redox potential of Cu[I]-species and Au[I]-species could gave some insight to the reaction mechanisms, so some other explanations for the interesting dual cascade cyclization was needed.

- I fully agree with the reviewer that the redox potential of Cu[I]-species and Au[I]-species may affect the cascade cyclization process, but this speculation is difficult to confirm especially when no valence change of metal is involved in this process. Further studies to elucidate it will be pursued in our laboratory.

7. Response to comment (reviewer 2): The authors adopted different basis sets (LanL2DZ for Cu and SDD for Au) for different transition metals in the same reaction system. However, in the computations, different basis sets will lead to different values of energy barriers. Therefore, it is not reliable to explain the much higher efficiency of the Au-involved reaction by using the much lower overall barrier height (OBH) with the Au catalyst relative to that with the Cu catalyst. Moreover, it is not convincing to adopt LanL2DZ basis set for Cu based on cost-effective consideration, since the atomic number of copper is smaller than that of gold. Therefore, I suggest the authors use SDD basis set for both copper and gold to perform the computations.

- We agree that different basis sets will lead to different values of energy barriers. However, we should mention that both LanL2DZ and SDD basis sets for Cu have been proven to be appropriate for the mechanistic investigations of Cu-based systems. Indeed, we have examined computationally both LanL2DZ and SDD basis sets for Cu on a series of key structures (precursor A, transition states TS_B , TS_C , TS_B' and TS_C') in the Cu-catalyzed reactions and listed the computed relative free energies in the Supplementary Table 4 of SI. The computations disclosed that the energy barriers given by two different basis sets of Cu agree well with each other.

廈門大學

XIAMEN UNIVERSITY

Xiamen, Fujian, China

8. Response to comment (reviewer 2): In the computational methods part of the Supporting Information, the sentence “The geometries of all the species were fully optimized by using density functional theory (DFT) of the M06 method” is not properly expressed, since M06 is one of the functionals of the density functional theory (DFT).

- The remark is revised as “The geometries of all the species were fully optimized by using density functional theory (DFT) method with the M06 functional”.

9. Response to comment (reviewer 3):

Comments:

Ye, Lu and co-workers report an interesting and important work on the divergent pathway of Cu- and gold-catalyzed cascade cyclization of azido-diyne. Using the same starting materials, the Cu-catalyzed oxidative transformation led to pyrrolo-quinolin-1-ones, while the gold-catalyzed cascade cyclization gave pyrrolo-indoles. The authors also propose the plausible mechanisms for the transformations based on the several control experiments and DFT computations.

Although the results are interesting and important, the manuscript needs to undergo a minor revision before it can be recommended for acceptance in Nat.Comm.

Following comments need to be addressed:

The divergent pathway in Cu- and gold-catalyzed such transformation is quite interesting. However, for the reaction condition of Au/oxidant-involving and Cu/oxidant-free, the authors only investigate the reactivity of substrate **1a** having a phenyl group. How about the reactivity of the substrates where the phenyl group in **1a** is replaced by an alkyl group, such as **1j**, under the condition of Au/oxidant-involving or Cu/oxidant-free? The authors should try the reactions, which maybe is helpful to better understand the mechanism.

- We have actually tried the ynamide **1j** under these conditions, as shown below. It was found that neither **2j** nor **3j** was formed in case of Au/oxidant condition or Cu/oxidant-free condition (**1j** was decomposed and the reaction gave a complicated mixture of products).

廈門大學
XIAMEN UNIVERSITY
Xiamen, Fujian, China

10. Response to comment (reviewer 3): Line 87~93: The authors should describe the reactions with mention of the catalyst involved in these reactions. This would improve the clarity for the readers.

- We have made some corrections according to the reviewer's kind suggestion.

I hope my responses to the reviewers' comments as well as all the changes made to the manuscript are satisfactory to your office.

Once again, thank you very much for your help with our manuscript.

Sincerely yours,

Longwu Ye, Ph. D.
Professor of Chemistry
College of Chemistry and Chemical Engineering
Xiamen University
Xiamen, Fujian 361005, China
<http://chem.xmu.edu.cn/group/lwye/index.html>

Reviewer #1 (Remarks to the Author):

I am reviewer 1 and I am pleased my previous four questions have been answered. This manuscript is now acceptable to me without any alterations.

Reviewer #2 (Remarks to the Author):

The authors have addressed the issues from the reviewers, and the revised version become suitable for publication.

Reviewer #3 (Remarks to the Author):

Ye, Lu and co-workers have addressed in the revised version all the issues raised by the reviewer during the previous submission. The revised manuscript is recommended for publication in Nature Communications.